# A scalable online tool for quantitative social network assessment reveals potentially modifiable social environmental risks

Amar Dhand [1,2], Charles C. White[3], Catherine Johnson[4], Zongqi Xia[5] & Philip L. De Jager [3,4]

Social networks are conduits of support, information, and health behavior flows. Existing measures of social networks used in clinical research are typically summative scales of social support or artificially truncated networks of ≤ 5 people. Here, we introduce a quantitative social network assessment tool on a secure open-source web platform, readily deployable in large-scale clinical studies. The tool maps an individual's personal network, including specific persons, their relationships to each other, and their health habits. To demonstrate utility, we used the tool to measure the social networks of 1493 persons at risk of multiple sclerosis. We examined each person's social network in relation to self-reported neurological disability. We found that the characteristics of persons surrounding the participant, such as negative health behaviors, were strongly associated with the individual's functional disability. This quantitative assessment reveals the key elements of individuals' social environments that could be targeted in clinical trials.

[1] Department of Neurology, Brigham and Women's Hospital, Harvard Medical School, Boston 02115 MA, USA. [2] Network Science Institute, Northeastern University, Boston 02115 MA, USA. [3] Broad Institute, Program in Medical and Population Genetics, Cambridge 02142 MA, USA. [4] Multiple Sclerosis Center and the Center for Translational & Computational Neuroimmunology, Department of Neurology, Columbia University Medical Center, New York 10032 NY, USA. [5] Department of Neurology, University of Pittsburgh, Pittsburgh 15260 PA, USA. Correspondence and requests for materials should be addressed to A.D. (email: adhand@bwh.harvard.edu)

Social connectivity is known to impact health. Social isolation is a predictor of mortality comparable to smoking, hypertension, and physical inactivity[1]. Social enrichment has a strong positive effect on biological[2] and functional health outcomes[3,4]. Social connections are also potentially modifiable, making them ideal targets for changing habits such as smoking, exercise, and diet[5].

Despite their promise in health, social networks are poorly understood in patient populations, and interventions aimed at networks are nascent. One main reason is a lack of clear definition of the network surrounding a patient[6,7]. Traditional social network metrics are actually summary indices of social support that query the total number of social contacts, social resources available, and community engagement[8]. Multiple clinical trials that have used such measures in patient populations have failed to demonstrate a change in patient outcomes[9–11]. A more precise set of measures are needed to map the specific people in the social system, one-by-one, and the nature of ties between persons to clarify a network's properties.

In this study, we introduce a social network assessment tool that quantifies patients' personal network structure and health characteristics in a web-based, secure, and scalable form. The tool is a survey adapted from a validated instrument, the General Social Survey[12], and captures the structure of social ties and composition of demographics and habits around the index patient. We demonstrate the utility of the tool by quantifying the personal networks of 1493 individuals at risk for multiple sclerosis. The participants are enrolled in the Genes and Environment in Multiple Sclerosis (GEMS) project, a prospective cohort study of people with first-degree family history of MS[13]. The goal of the GEMS project is to identify novel genetic and environmental risk factors, including the social environment. Prior work has shown that asymptomatic MS family members who have a high burden of genetic and environmental risk factors have evidence of diminished neurologic function[14]. Here, we show a relationship in the GEMS cohort between social network metrics and neurological disability. We demonstrate that quantifying social networks in large-scale clinical studies offers an effective platform to identify previously unknown social environment risk factors that are potentially modifiable.

## Results

**Creating a scalable online tool to assess social networks.** We designed a HIPAA-compliant structured social network questionnaire adapted primarily from the General Social Survey[12,15] (Supplementary Methods 1). The schema of data acquisition and potential use is presented in Fig. 1. The questionnaire comprises ~48 questions with adaptation to responses. The estimated completion time of the questionnaire is 10–15 min. The questionnaire begins with three traditional name generators, in which participants named all people with whom they had discussed important matters, socialized, or sought support in the last 3 months. The number of people who could be named was not capped. Next, participants answered questions that evaluate the connections between each pair of the first ten persons in the network, including the strength of ties in three levels (strangers, weak, and strong). Finally, participants answered questions about the characteristics and health habits of each of the first ten persons in the network[7]. The online questionnaire was hosted on the Research Electronic Data Capture (REDCap) server, a secure web platform for administering questionnaires in clinical research[16]. A version of the instrument is available for use in the REDCap Shared Library. Code to analyze and visualize data created from the instrument is available on GitHub.

The assessment generated two main categories of network metrics, structure, and composition, based on graph theoretical statistics. Within the category of social network structure, size is the number of individuals in the network, excluding the index participant or "ego". Density is a measure of connectivity of individuals in the network, calculated as the sum of ties, excluding the ego's ties, divided by all possible ties[17]. Constraint is a more detailed version of density that quantifies the extent to which the ego's connections are to individuals who are connected to one another. Effective size is the number of non-redundant members in the network[18]. Maximum degree is the highest number of ties by a network member, and mean degree is the average number of ties by a network member. Equations for these measures are available in Supplementary Methods 2.

Within the social network composition category, several metrics quantify the ratio of member characteristics in the network. For instance, the percent kin is the percent of individuals in the network who are family members. Standard deviation of age represents the range of ages. The diversity of sex index is the mix of men and women in the network, according to the index of qualitative variation[19], with a value of 1 indicating equal mix of men and women. The diversity of race is the mix of races similarly calculated. Importantly, the questionnaire also queries the health behavior environment around the participant by examining the percentage of the network members with negative health habits, including smoking, sedentary lifestyle, not visiting doctors regularly, and poor compliance of prescription medications. All compositional variables were created to account for network size. Specifically, the number that fits a category was divided by the total size to create the percentage.

**Demonstrating network quantification in a nation-wide cohort.** We assessed the social networks of 1493 GEMS participants from across the United States (Supplementary Fig. 1), which represented 57% of the cohort as of October 2016. In Table 1, we report the demographic and clinical information of the cohort at the time of the study, separated into subgroups of asymptomatic participants and participants with an MS diagnosis. Asymptomatic participants had a lower age on average than participants with an MS diagnosis, consistent with the previously reported demographics of the cohort[13].

The primary outcome measure of functional disability was the MSRS-R, a self-reported outcome of functional disability validated for people with MS. The MSRS-R is a brief questionnaire that correlates with traditional clinical instruments[20,21]. The eight domains of MSRS-R include walking, using arms and hands, vision, speaking clearly, swallowing, cognition, sensation, and the bowel and bladder function for a maximum score of 32. In this cohort of primarily asymptomatic people at risk for MS, we chose MSRS-R as an outcome measure because few alternative self-reported outcome measures have the advantages of being concise and validated in early MS. As expected, the median MSRS-R score was higher on average in the MS group than in the asymptomatic group.

To visualize each participant's social network structure, we plotted a montage of all participants' networks, ranging from the smallest to the largest, with the strength of each tie highlighted in color (Fig. 2). The average network consisted of eight people who were densely linked (67% of all possible ties were present). Furthermore, an average of 44% of all network members were kin, 38% were supportive of the index participant, and there was a nearly equal mix of men and women (diversity score of 0.89 with one being an equal mixture of men and women). Race, on the other hand, was not varied within networks with a diversity score of 0, indicating that most members in a participant's network

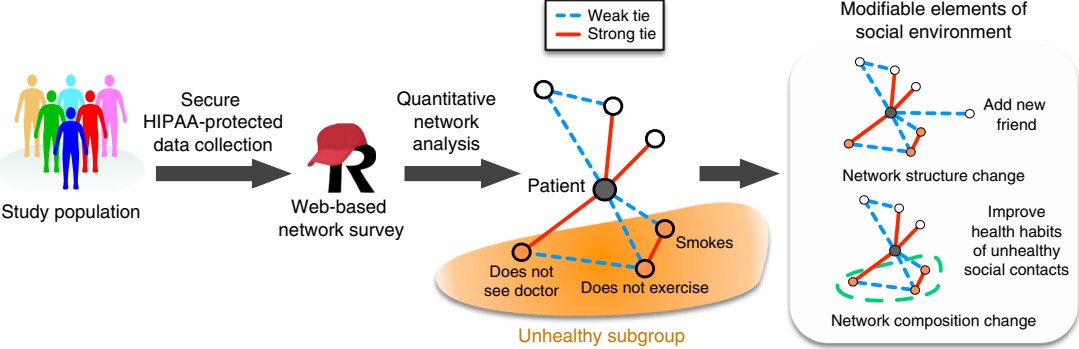

**Fig. 1** Overview of data collection, analysis, and interventions. This flowchart shows the social network data acquisition, identification of modifiable elements in the social environment, and potential intervention strategies

| Table 1 Demographics and clinical characteristics of the participants | | | |
|---|---|---|---|
| **Characteristic** | **Asymptomatic** $n = 1378$ | **MS** $n = 115$ | **P-value**[a] |
| Age, mean (SD), y | 37.85 (8.34) | 43.14 (7.60) | < 0.001 |
| Male sex, no. (%) | 269 (19.5) | 19 (16.5) | 0.51 |
| Years of education, median [IQR] | 16 [16,18] | 16 [15,18] | 0.18 |
| Married, no. (%) | 914 (66.7) | 86 (76.1) | 0.051 |
| Living alone, no. (%) | 198 (13.4) | 12 (10.4) | 0.45 |
| Age of onset of MS symptoms, mean (SD) | NA | 30.50 (8.70) | NA |
| Age of diagnosis of MS, mean (SD) | NA | 34.36 (7.74) | NA |
| MSRS-R, median [IQR] | 1.00 [1.00, 2.00] | 7.00 [3.00, 11.00] | < 0.001 |

[a]P-values calculated using t test for age; chi-squared test for female sex, married, and living alone; and Wilcoxon signed-rank test for years of education and MS rating scale score-revised (MSRS-R)

were of the same race. Weak ties, denoting those who are less familiar with the participant, ranged from 20% to 67% depending on the measure. The percent of individuals who were known for less than 6 years by the respondent was 20% in asymptomatic persons and 12% in MS patients ($P = 0.001$, Wilcoxon signed-rank test), suggesting a reduction in recent acquaintances in participants with an MS diagnosis. Otherwise, differences in network structure and general network composition between asymptomatic and MS participants were small and not significant (Table 2).

To visualize the milieu of health habits around the participant, we plotted a montage of all participants' networks, ranging from the healthiest environment to the least healthy (Fig. 3). On average, the network composition with respect to health habits skewed toward social environments in which most network members have healthy habits. Seventeen percent of participants had personal networks in which all members were healthy. On average, the percent of network members who do not exercise was 33%, and this was the highest value out of the examined negative health habits. There was a weak negative correlation between network size and the percentage of network members with unhealthy habits (Pearson's correlation $= -0.13 \pm 0.05$, $P < 0.0001$). Because we did not detect differences in network composition with respect to healthy habits between asymptomatic and MS participants, we were able to pursue joint analyses of these two subgroups.

Having established the basic properties of our data, we examined the relationship between network metrics and self-reported functional disability outcome. Given the number of network metrics and to account for multiple testing burdens, we grouped the network variables into structure and composition categories. We then used a permutation-based omnibus test to examine the associations of these two groups of network metrics with the MSRS-R. The observed distribution of P-values in the

omnibus test was greater than chance for network composition ($P = < 0.0001$, all; $P = 0.008$, asymptomatic subgroup; $P = 0.001$, MS subgroup), but not for network structure ($P = 0.066$, all; $P = 0.14$, asymptomatic subgroup; $P = 0.25$, MS subgroup) (Table 3, Fig. 4). Thus, our global assessments indicated that network composition, rather than network structure, was associated with self-reported functional disability based on the MSRS-R scores (Table 3).

To deconstruct these global effects of the social network, we examined the association of individual network metrics with the MSRS-R, adjusting for sex, age, marital status, and years of education (Table 4). None of the network structure metrics were significantly associated with MSRS-R score, consistent with the global assessment. Two network composition features were significantly associated with MSRS-R score: the percent of network members who (1) do not go to a doctor regularly or (2) are deemed to have a negative health influence on the respondent. The strongest association was with the percent of network members who are deemed to have a negative health influence ($\beta = 0.017 \pm 0.005$, $P = 0.016$, linear regression).

In exploratory analyses, we examined the relationship between each individual's Genetic and Environmental Risk Score (GERS) and her or his social network size. The GERS is an aggregate estimate of an individual's MS risk based on validated genetic and environmental susceptibility factors. We have previously reported that the GERS is informative of MS risk beyond family history in the GEMS cohort of first-degree family members[13]. Using the published GERS based on previously reported genetic and environmental risk factor data available among a subset of the GEMS participants ($n = 999$ all, $n = 920$ asymptomatic subgroup, $n = 79$ MS subgroup), we noted an association in linear regression between larger network size and increased GERS ($\beta = 0.82 \pm 0.19$, $P = 2.43 \times 10^{-5}$, all) (Supplementary Table 1). This finding appears to be driven by the larger network size of women

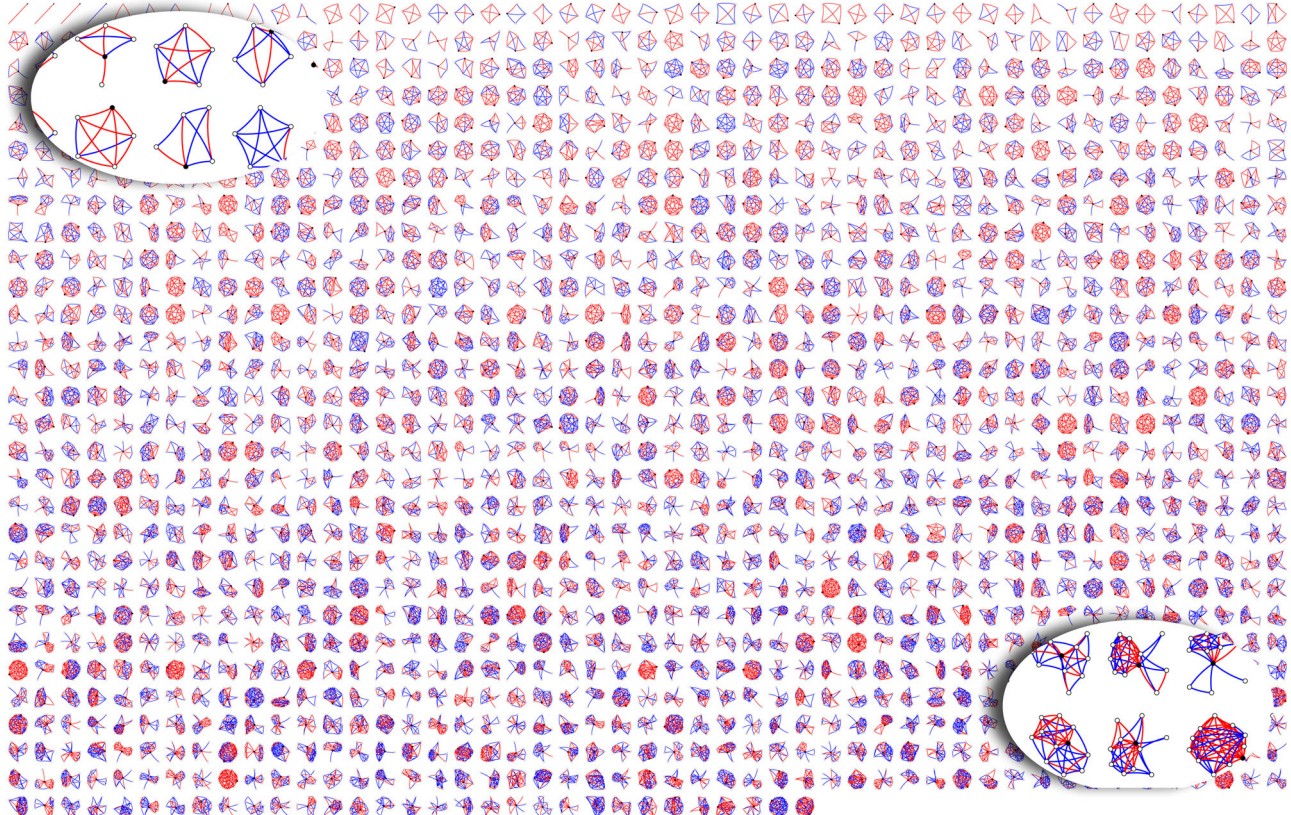

**Fig. 2** Structure of participants' personal social network. Each small network has a black circle that represents the participant who is surrounded by white circles who are the network members. The lines connecting the circles are red if the relationship is strong and blue if the relationship is weak. Networks are arranged from the smallest (top left) to the largest (bottom right)

participants relative to men. In a regression analysis, network size is inversely related to male sex ($\beta = -1.87 \pm 0.42$, $P = 8.71 \times 10^{-6}$, all). Among asymptomatic participants, both a history of mononucleosis ($\beta = 1.13 \pm 0.40$, $P = 0.005$) and a higher genetic risk score for MS susceptibility ($\beta = 0.65 \pm 0.24$, $P = 0.006$) were also associated with a larger network size in the linear regression (Supplementary Table 1).

**Discussion**

In this in-depth analysis of social networks in family members of MS patients, we demonstrate the ease and utility of deploying our online questionnaire that evaluates an individual's social network in a structured manner. In a few weeks and using only electronic communication, we collected complete data on 1493 individual GEMS participants. This large data set allowed us to pursue analyses in a statistically robust manner and to produce highly significant results. These results represent an important milestone in studies of MS and other neurologic conditions with a long prodromal neurodegenerative phase by providing investigators with the key data needed to support power calculations and guide future study designs. In particular, we found that asymptomatic family members at risk of MS have enough variance in our measure of self-reported disability to yield strong association results with compositional but not structural variables. Most prominently, the health habits of persons in their social environment was strongly associated with the participant's self-reported neurological dysfunction, and the percent of network members who have a negative health influence had the strongest association with disability. While these results need to be validated, they show (1) that studies of "at risk" individuals in which overt symptoms of a neurologic disease have not yet become

manifest are feasible and (2) that network composition is an area that deserves further dissection in individuals at risk for MS and perhaps for other neurodegenerative diseases.

Our assessment adds to a growing list of web-based personal network surveys that translate the complexity and burdensome features of this type of questionnaire into a more usable and scalable form[22]. Two examples in public health include: (1) EgoWeb 2.0[23], an open-source software that may be used for motivational interviewing using network graphics and (2) OpenEddi[24], a tool designed for interactive, tablet, or mobile-ready field collection of network data. Our tool is unique, in that it is a HIPAA-compliant data collection tool, able to be completed by patients without an interviewer, and has the capability to handle large volumes of data from clinical populations using electronic communications. The assessment also included questions customized for patients or at-risk individuals with a focus on social support and health-related behaviors of network members. These dimensions are critical for future planning of network interventions to improve health and quality-of-life outcomes in clinical settings.

One mechanism that may explain some of our findings is the tendency of individuals to associate with others who are similar to themselves or homophily. Similarity-breeding social connection has been described in other social network studies[25]. Race and ethnicity are the strongest linkage factors leading to homogenous personal environments[25], and we found this in our study as well. However, there are many examples of health behavior homophily. Children's social network composition is significantly associated with several aspects of children's own health[26]. Latrine ownership in rural India is correlated with latrine usage among social contacts, after control of caste, education, and income[27]. An

**Table 2 Network characteristics**

| Characteristic | Asymptomatic, $n = 1378$ | MS, $n = 115$ | P-value[a] |
|---|---|---|---|
| *Network structure*[b] | | | |
| Size, median [IQR] | 8.00 [6.00, 12.00] | 8.00 [5.00, 11.50] | 0.130 |
| Density, median [IQR] | 67.00 [50.00, 89.00] | 69.00 [53.00, 90.00] | 0.170 |
| Constraint, median [IQR] | 44.00 [37.72, 53.03] | 44.71 [38.19, 56.17] | 0.315 |
| Effective size, median [IQR] | 4.00 [2.80, 5.25] | 3.55 [2.50, 5.07] | 0.053 |
| Maximum degree, median [IQR] | 5.00 [4.00, 7.00] | 5.00 [4.00, 8.00] | 0.987 |
| Mean degree, median [IQR] | 4.00 [2.80, 5.00] | 4.00 [2.50, 5.40] | 0.493 |
| *Network Composition–General*[c] | | | |
| Percent kin, median [IQR] | 43 [30, 62] | 50 [33,67] | 0.205 |
| Percent who are supportive, median [IQR] | 38 [25, 50] | 40 [21,50] | 0.561 |
| Standard deviation of age, median [IQR] | 12.76 [10.04, 15.38] | 12.98 [10.54, 16.89] | 0.161 |
| Diversity of sex, median [IQR] | 0.89 [0.64, 0.96] | 0.82 [0.64, 0.96] | 0.108 |
| Diversity of race, median, Percentile {90th, 95th, 99th,100th}[d] | 0 {0.44, 0.55, 0.72, 1.20} | 0 {0.41, 0.59, 0.77, 0.77} | 0.046 |
| Percent contacted weekly or less often, median [IQR] | 67 [50, 80] | 67 [45, 80] | 0.896 |
| Percent who have been known for less than 6 years, median [IQR] | 20 [0, 43] | 12 [0, 33] | 0.001 |
| Percent who live more than 15 miles away, median [IQR] | 33 [17, 50] | 33 [20, 56] | 0.514 |
| *Network Composition–Health Habits*[e] | | | |
| Percent who smoke, median [IQR] | 0 [0, 20] | 0 [0, 40] | 0.164 |
| Percent who do not exercise, median [IQR] | 33 [14, 54] | 25 [10, 50] | 0.068 |
| Percent who do not take medications regularly, median, Percentile {90th, 95th, 99th,100th} | 0 {0, 14, 33, 100} | 0 {0, 17, 24, 50} | 0.709 |
| Percent who do not go to doctor's appointments, median, Percentile {90th, 95th, 99th,100th} | 0 {0, 12, 25, 100} | 0 {0, 15, 48, 100} | 0.314 |
| Percent who have a negative influence on health, median, Percentile {90th, 95th, 99th,100th} | 0 {29, 46, 71, 100} | 0 {20, 33, 78, 100} | 0.150 |

[a]P-values calculated using Wilcoxon signed-rank test
[b]Network structure is quantified into graph theoretic statistics. See definitions in Methods
[c]Network composition–General is the range of characteristics of people around the participant. See definitions in Methods
[d]Percentile is used to better understand the right-skewed distribution of the variables of race and certain health habits
[e]Network composition–Health Behavior is the range of health habits of people around the participant

individual's weight is influenced by obesity of spouses and same-sex social contacts[28], and incident type 2 diabetes is associated with obesity in spouses[29]. Aspirin use is correlated with aspirin use among friends and family[30]. Taken together, these findings point to core human behaviors that are shared among like-minded social contacts, with eating and physical activity as major driving forces for these effects.

Two more mechanisms that may explain the association of network members' health habits and the participant's neurological disability are social contagion and antecedent exposures. Social contagion is a type of social influence in which behavior in one or many network members affects the behavior of the index participant. Detection of this effect requires longitudinal data and network modeling, such as stochastic actor-oriented or instrumental variable approaches, to understand the spread of behaviors through social ties. For example, one study shows the spread of physical activity in 1 million users of a smartphone running application[31]. Antecedent exposures influencing both parties may be another contributor. For example, rural environments with poor access to medical services may influence the habits of all members of the network with regard to seeking medical care. Finally, a combination of these factors may explain the association of poor health habits in the network and a person's neurological disability.

The association between an individual's susceptibility for MS, as determined by GERS, and social network size is a preliminary finding that requires further investigation. This may be explained by the inclusion of sex as a component of GERS[13] and prior observation that women tend to have larger social networks[15]. However, the imbalance of men (19%) and women (81%) in this study potentially complicates the interpretation. Another explanation is that larger network size reflects broader exposure to infectious agents that are associated with MS susceptibility, such as history of infectious mononucleosis[13]. Indeed, we observed a positive association between mononucleosis and network size among asymptomatic participants. Finally, the role of genetic factors in network size is provocative, but the effect is modest and needs further investigation.

Our study has limitations. First, we were unable to establish causality and directionality of the associations or the mechanisms of homophily in this cross-sectional study. Within the GEMS platform, we are gathering longitudinal social network data. Second, the primary outcome measure of neurological disability (MSRS-R) was skewed toward low scores due to the larger proportion of self-reported asymptomatic participants in the GEMS cohort who have low scores in this instrument. This could reduce the precision of our analyses due to a floor effect. Further, the study may be underpowered to compare asymptomatic and MS subgroups, given the modest number of the MS cases (i.e., familial MS). Larger studies of individuals with sporadic MS will better answer whether social network variables influence disease worsening in MS. Third, unmeasured confounders that influence report of social networks and functional disability could have affected our findings. We attempted to address this limitation by adjusting for major factors reported in the literature, including age, sex, and marital status. Fourth, we ascertained social network metrics based on participants' self-report of their social networks. While this approach may introduce unknown biases, prior work reassuringly had shown self-reported personal networks of intimate contacts to be accurate[32]. Finally, this study of the GEMS

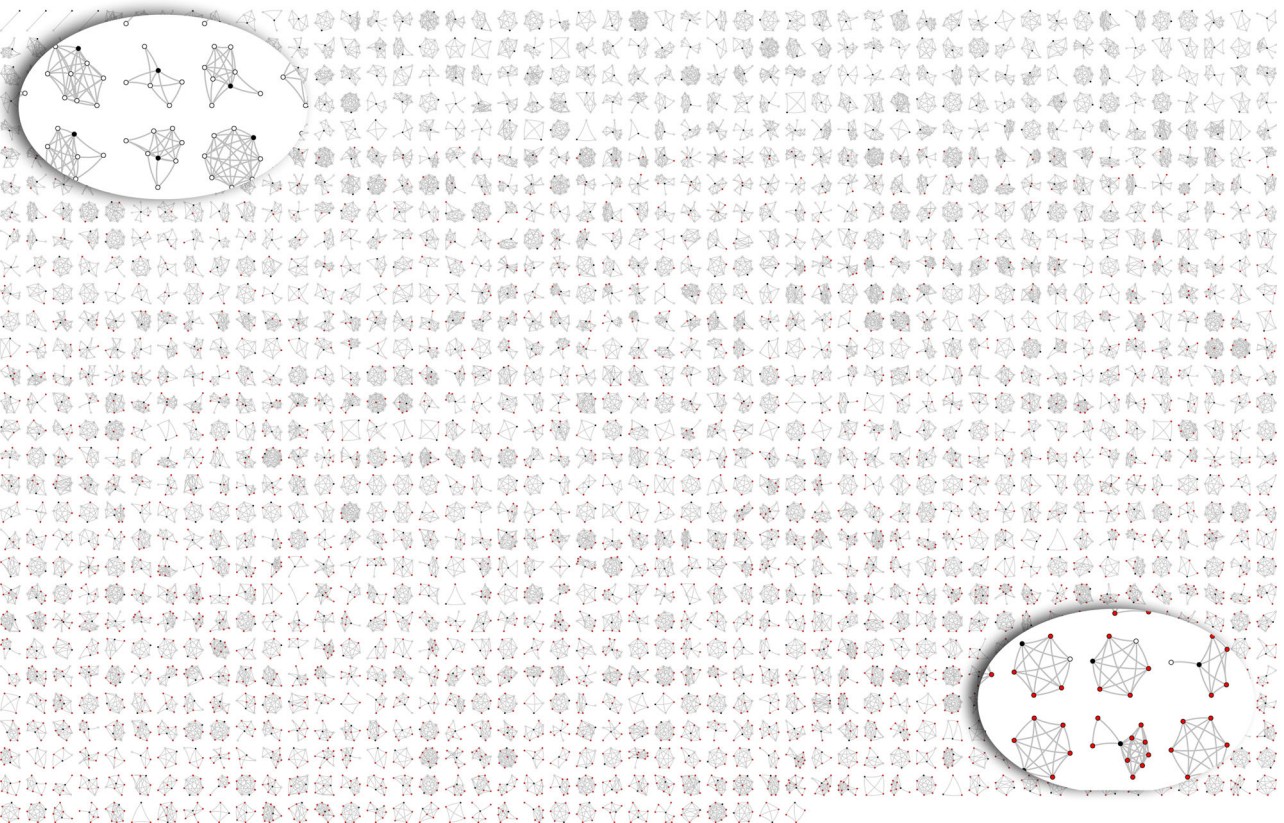

**Fig. 3** Health habits in participants' personal social network. In each network, a black circle is the participant, a white circle is a healthy social contact, and a red dot is an unhealthy social contact. Unhealthiness is defined as someone who does any of the following: smokes, does not exercise, does not visit doctors regularly, or not compliant with prescription medications. Networks are arranged from least negative health influence (top left) to most negative health influence (bottom right)

| Table 3 Relationship of the composite categories of network variables to MSRS in all participants | | | | | |
|---|---|---|---|---|---|
| **Variable category** | **Number of variables** | **Top variable** | **Top variable P-value** | **Top variable FDR value** | **Composite P-value[a]** |
| Structure | 6 | Total size | 0.025 | 0.133 | 0.066 |
| Composition | 13 | Percent who do not go to doctor's appointments | $7.4 \times 10^{-8}$ | $9.6 \times 10^{-7}$ | < 0.0001 |

FDR false discovery rate
[a]Permutation-based omnibus test is described in the methods

participants, who were recruited through advocacy groups, social media, and electronic communications, may not have broad generalizability because these participants are more socially engaged and better educated than the general population. Future studies of more diverse populations and other chronic neurological disorders will be critical.

The social environment is ubiquitous and important for understanding human disease etiologies and outcomes. Social network features, in general, represent an emerging group of metrics that inform aspects of health and disease, but are not currently well captured by many biomedical research studies. We outline an approach of quantitative social network analysis that is readily adaptable in clinical investigations. The questionnaire that we have developed for quantifying social networks is available through the open-source REDCap platform. In the empirical work described, we found that the health behaviors of persons surrounding an individual at risk for MS were associated with the individual's own functional status. These results suggest that interventions aimed at modulating network composition through education or treatment of members in a social network holds the promise of a novel complementary approach to managing MS onset and disease course.

## Methods

**Study design and participants.** In a cross-sectional design, we invited GEMS participants to complete an online questionnaire assessing social networks and current neurological disability in October 2016 (Supplementary Methods 1). The questionnaire was live for 6 weeks, with reminders sent to non-responders. At the time, the GEMS cohort included 2632 first-degree family members from across the United States recruited using patient advocacy groups, social media, and word-of-mouth[13]. The inclusion criteria were: being 18 to 50 years of age at enrollment and having at least one first-degree relative with a diagnosis of MS (e.g., parent, full-sibling, or child). While asymptomatic family members who are at risk for MS represent the main focus of the GEMS project, we also recruited family members who already have a MS diagnosis for comparison in this cross-sectional study. MS cases were confirmed by review of medical records. The institutional review boards

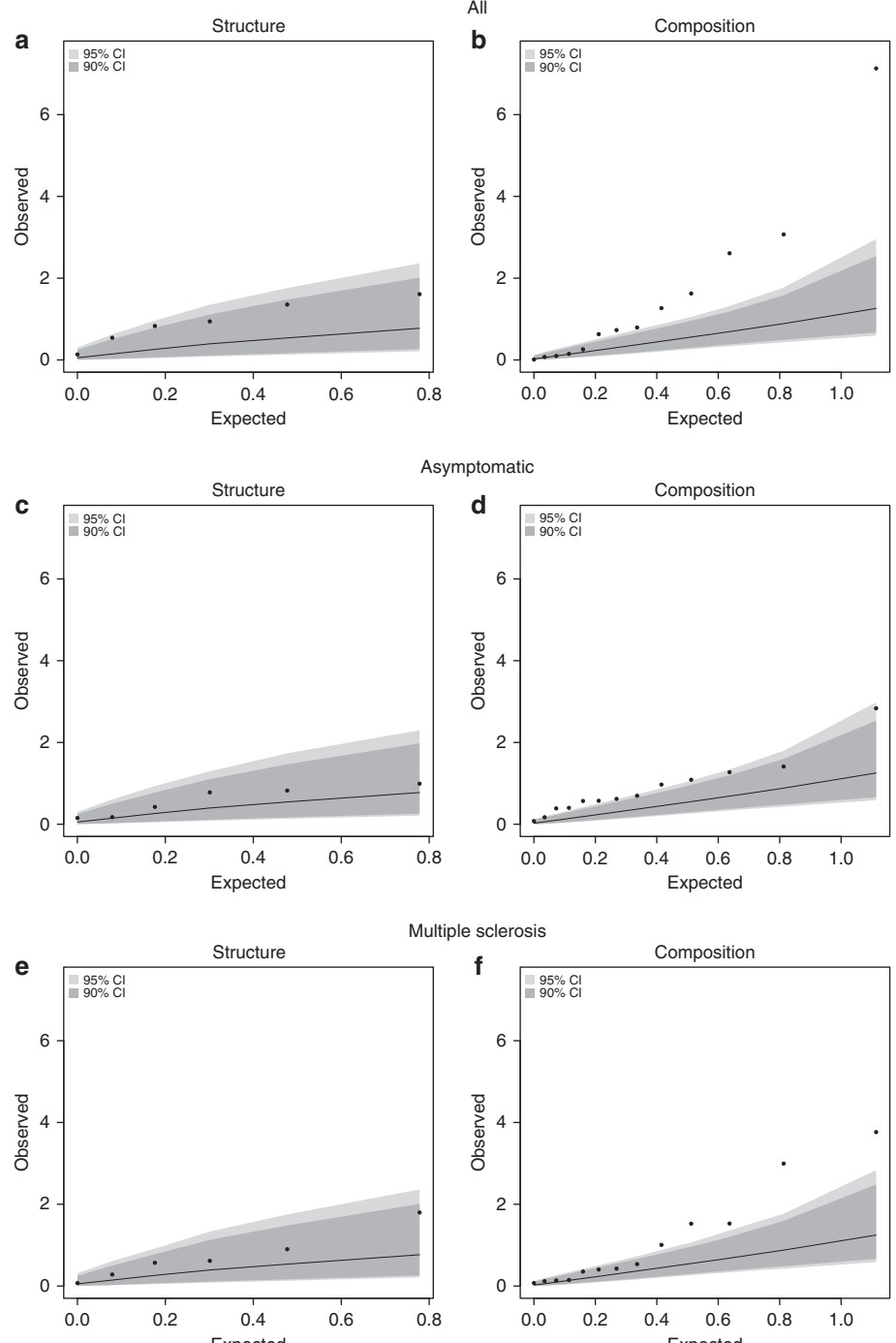

**Fig. 4** Comparison of expected versus observed regression results. Quantile–quantile plot of expected versus observed *P*-values of composite network structure and network composition metrics in relation to neurological function and disability in the full cohort (**a**, **b**) and subgroups of asymptomatic (**c**, **d**) and MS participants (**e**, **f**). The expected *P*-values (-log10[P-value]) are shown on the x-axis and the observed *P*-values (-log10[P-value]) are shown on the y-axis. The dark gray area indicate the confidence interval ranges as generated by chance at a threshold of *P* = 0.10 and the light gray is for *P* = 0.05. The observed values for composition, and not structure, are outside of the gray areas, suggesting that composition is associated with the MSRS-R score beyond chance after accounting for multiple testing burden and correlation structure of the composition variables

of all participating sites (Partners HealthCare, National Institutes of Health, and University of Pittsburgh) approved the study. All participants provided written informed consent.

**Statistical methods**. To compare the demographic characteristics between asymptomatic participants and confirmed MS cases, we performed a *t* test for age, chi-squared tests for dichotomous variables of sex, marital status, and living alone,

as well as non-parametric Wilcoxon rank-sum tests for years of education and MSRS-R. Similarly, we performed non-parametric Wilcoxon rank-sum tests to compare network metrics between asymptomatic participants and participants with MS diagnosis.

To assess the association with MSRS-R score, we performed a linear regression for each network variable, adjusting for age, sex, and marital status. In this analysis, MSRS-R was modeled as the dependent variable and each network characteristic as the independent variable. Within each network metrics category (structure and

**Table 4 Relationship of individual network variables to MSRS-R**

| Variable | $\beta$ | Standard error | Adjusted $P$-value[a] | FDR[b] |
|---|---|---|---|---|
| *Network structure* | | | | |
| Size | −0.025 | 0.013 | 0.052 | 0.197 |
| Density | 0.007 | 0.365 | 0.985 | 0.985 |
| Constraint | 0.004 | 0.007 | 0.537 | 0.729 |
| Effective size | −0.035 | 0.05 | 0.487 | 0.712 |
| Maximum degree | −0.041 | 0.044 | 0.347 | 0.564 |
| Mean degree | 0.003 | 0.052 | 0.958 | 0.985 |
| *Network Composition–General* | | | | |
| Percent kin | 0.001 | 0.004 | 0.769 | 0.876 |
| Percent who are supportive | −0.005 | 0.004 | 0.198 | 0.47 |
| Standard deviation of age | −0.006 | 0.017 | 0.701 | 0.876 |
| Diversity of sex | −0.332 | 0.359 | 0.356 | 0.564 |
| Diversity of race | 0.686 | 0.423 | 0.105 | 0.333 |
| Percent contacted weekly or less often | −0.009 | 0.004 | 0.023 | 0.147 |
| Percent who have been known for less than 6 years | 0.001 | 0.004 | 0.784 | 0.876 |
| Percent who live more than 15 miles away | −0.003 | 0.004 | 0.346 | 0.564 |
| *Network Composition–Health* | | | | |
| Percent who smoke | 0.006 | 0.003 | 0.045 | 0.197 |
| Percent who do not exercise | −0.003 | 0.003 | 0.296 | 0.564 |
| Percent who do not take medications regularly | 0.018 | 0.012 | 0.123 | 0.334 |
| Percent who do not go to doctor's appointments | 0.045 | 0.015 | 0.002 | 0.023 |
| Percent who have a negative influence on health | 0.017 | 0.005 | 0.001 | 0.016 |

[a]Adjusted for potential confounders, sex, age, marital status, and years of education via linear regression as described in the Methods
[b]FDR is false discovery rate, controlling for multiple testing

composition), we calculated the false discovery rate to adjust for multiple testing. To examine any potential bias due to non-normal distributions, we performed a sensitivity analysis applying non-parametric spearman correlation tests.

To examine the hypothesis that as a category, social network variables were associated with the MSRS-R score, we performed an empirical omnibus test. In the first stage of this analysis, we calculated the $P$-values of association between each network variable and MSRS-R score using linear regression as described above. In the second stage, we used a Fisher's meta-analysis to combine these $P$-values and calculate a chi-squared statistic. We then compared this chi-squared statistic to an empirical distribution of chi-squared statistics as generated by 10,000 random permutations. By permuting the MSRS-R score, we maintained the correlation structure of the network variables. The empirical omnibus $P$-value was then calculated as the number of times that the chi-squared statistic from the 10,000 permutations was greater than the true chi-squared statistic divided by the total number of permutations. To generate a quantile–quantile plot, we plotted the observed −log10 ($P$-value) of each pair of association between a network variable and MSRS-R score against the expected −log10 ($P$-value). The 90th and 95th empirical confidence intervals were determined using empirical $P$-values as generated by the 10,000 permutations. We performed the omnibus test in all participants as well as in the subset of asymptomatic participants and the subset of participants with MS diagnosis.

In exploratory analyses, we assessed the relationship of GERS (a published estimate of MS risk based on an individual's known genetic burden and environmental exposures for MS susceptibility) and social network metrics. Here, we performed linear regressions adjusting for age, modeling network size as the dependent variable, and the GERS (and its components: history of infectious mononucleosis, sex, smoking status environmental risk score, and genetic risk score) as the independent variables. All analyses were performed in R version 3.2[33]. All statistical tests were two-sided. Given the exploratory nature of the analysis and data, power calculations were not performed prior to analysis. Permutations and nonparametric tests were used to avoid bias due to any non-normal data or unequal variances between groups, as necessary.

**Code availability**. An updated version of the instrument called "Personal Network Survey for Clinical Research" is available in the REDCap Shared Library. We have also uploaded a comprehensive R codebase for researchers who use the instrument to analyze and visualize their data available at: https://github.com/AmarDhand/PersonalNetworks. R code used specifically for this project can be made available upon request.

## Data availability

The data used in this study is freely available as a supplement to this manuscript (Supplementary Database 1).

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

## Acknowledgements

This work was supported by NIH grants K23HD083489, K08NS079493, the Football Players Health Study at Harvard University, and National Multiple Sclerosis Society RG-5003-A-2. We acknowledge Angela H. Kim for developing the figures. The Football Players Health Study is funded by a grant from the National Football League Players Association. The content is solely the responsibility of the authors and does not necessarily represent the official views of Harvard Medical School, Harvard University or its affiliated academic health care centers, the National Football League Players Association, or Brigham and Women's Hospital.

## Author contributions

A.D., C.C.W., Z.X. and P.L.D. conceived the study. A.D., C.C.W. and C.J. collected the data. A.D., C.C.W., Z.X., C.J. and P.L.D. performed data analysis. A.D., C.C.W., Z.X., C.J. and P.L.D. wrote the paper.

## Additional information

**Competing interests:** The authors declare no competing interests.

