## [Peer Review File · Nature Communications]

Reviewers' Comments:

Reviewer #1:

Remarks to the Author:

Review for:

"A Scalable, Web-based, and Quantitative Social Network Assessment for Identifying Potentially Modifiable Risks in the Social Environment"

Amar Dhand, Charles C. White, Catherine Johnson, Zongqi Xia, Philip De Jager

In this paper, the authors present a new quantitative social network assessment tool on a secure open source web platform that could be helpful for large scale clinical studies. They use the tool for demonstration to quantify the social network structure and composition of 1493 persons in US with a risk to develop multiple sclerosis. While network structure measures were not associated with self-reported neurological disabilities, negative health habits of individual's surroundings strongly correlated with the neurological disorder.

There are several limitations of the current study including lack of causal conclusions, self-reporting of both network characteristics and neurological functions and in terms of generalization of the results. On the other hand, the paper is well written, the study technically sounds and the topic is interesting in the broad area of designing interventions and evaluating clinical studies. For those reasons, even if I cannot be very enthusiastic, I believe an improved version could be accepted for publication in Nature Communications. I have some remarks that need to be addressed:

1. Figure 1 (the last part in the right) is not very informative. While the reader understands from the text the separation between the structure and composition, Figure 1 at the right get things complicated.
2. In lines 94-96 the authors state that "the maximum and mean degree ... are indicating the distribution of ties in the network." Usually social networks are complex in the sense that the degree distribution follows a heterogeneous distribution. Therefore, maximum and mean degree is not conclusive for tie distribution in a network. I propose that the authors change a bit the wording here.
3. Figure 2 is not informative and add nothing to the contribution of the study. It is a very simple scatter plot of the locations of individual participants that gets very complicated in large cities (circles are one over the other) – and it is difficult to fit in a Nature Communication paper. I would have this figure in the supplementary material rather than in the main text. It looks like that the location information is self-reported through an address. Maybe a heat map at the level of county or zip code would look much better. It looks like there is a large correlation between population of a city (or county) and the number of participants. This is something that the authors could report.
4. In line 142 the authors refer to Fig 3. It should be Fig 4. Also in the legend of Fig 4 it refers to "orange" color. I believe the unhealthy friends are red colored.
5. In Figure 3, it looks like that about ~25% of all ego's networks have healthy habits. I think is worth mentioning the exact percentage. Also, is the size of the network correlated (at least weakly) with how "unhealthy" is the network? It should be that the more the friends I have the largest the probability that at least one of them has unhealthy habits. Therefore, the way networks are ranked in Figure 3 is related to the way that networks in Figure 4 are ranked and visualized.
6. My above comment on the correlation between network size and percentage of friends that follow unhealthy habits is the main reason (from my point of view) behind the result that the authors report in Table 3, i.e. the association between the MSRS and the size of the network. There is this confounding here that the structure (at least the size) is highly correlated with the network composition - health. This is something that needs to be discussed in the paper – maybe in page 11.
7. At lines 232-247 the authors discuss about "homophily". It is a common sense that association (correlation) of health behaviors in a social network can be attributed to homophily (the tendency of similar individuals to connect in a social network) or to contagion effects, or to the fact that

certain areas of a network subject to same externalities. In line 234 the authors state that a "potential mechanism of homophily is social contagion." I would say that potential mechanism of the association in health behaviors are homophily and social contagion.

Reviewer #2:

Remarks to the Author:

This is a very well written and interesting paper that looks at a novel metric that may impact on health. As such it provides new information that may be used to influence the risk of developing MS in those at risk. The social network is certainly a very topical area for research and the authors address this in the GEMS cohort.

I have several questions.

Firstly the MSRS_R was used as the primary outcome measure of disability but was heavily skewed to a zero response. Did the authors consider using a hurdle model to analyse the data??

What was the influence of educational level on social networks could this be a confounder?

In table 2 it would be useful to provide the actual percentages who smoke etc in addition to the medians IQR.

Could the authors also comment on how representative their sample was of the whole GEMS cohort would it be more likely that those who respond have a greater or lesser social network than those who did not?

The information presented here would definitely allow me to reproduce this work.

Bruce Taylor

University of Tasmania

Response to the Referees' Comments

Reviewer #1 comments:

In this paper, the authors present a new quantitative social network assessment tool on a secure open source web platform that could be helpful for large scale clinical studies. They use the tool for demonstration to quantify the social network structure and composition of 1493 persons in US with a risk to develop multiple sclerosis. While network structure measures were not associated with self-reported neurological disabilities, negative health habits of individual's surroundings strongly correlated with the neurological disorder.

There are several limitations of the current study including lack of causal conclusions, self-reporting of both network characteristics and neurological functions and in terms of generalization of the results. On the other hand, the paper is well written, the study technically sounds and the topic is interesting in the broad area of designing interventions and evaluating clinical studies. For those reasons, even if I cannot be very enthusiastic, I believe an improved version could be accepted for publication in *Nature Communications*. I have some remarks that need to be addressed:

1. Figure 1 (the last part in the right) is not very informative. While the reader understands from the text the separation between the structure and composition, Figure 1 at the right get things complicated.

We have simplified the right side of Figure 1. We have removed two graphs and focused the reader on two modifications possible: adding a new friend or improving the habits of unhealthy social contacts.

2. In lines 94-96 the authors state that “the maximum and mean degree ... are indicating the distribution of ties in the network.” Usually social networks are complex in the sense that the degree distribution follows a heterogeneous distribution. Therefore, maximum and mean degree is not conclusive for tie distribution in a network. I propose that the authors change a bit the wording here.

We have removed this wording and changed the sentence as follows:

P.5, line 99: Maximum and mean degree are the network members who have the most number of ties and average number of ties, respectively.

3. Figure 2 is not informative and add nothing to the contribution of the study. It is a very simple scatter plot of the locations of individual participants that gets very complicated in large cities (circles are one over the other) – and it is difficult to fit in a Nature Communication paper. I would have this figure in the supplementary material rather than in the main text. It looks like that the location information is self-reported through an address. Maybe a heat map at the level of county or zip code would look much better. It looks like there is a large correlation between population of a city (or county) and the number of participants. This is something that the authors could report.

We have moved the figure to supplementary material. The location of participants is interesting but not directly related to the point of the paper, so we do not add these details into the manuscript. We will consider this variable for future analyses.

4. In line 142 the authors refer to Fig 3. It should be Fig 4. Also in the legend of Fig 4 it refers to “orange” color. I believe the unhealthy friends are red colored.

The figure numbers are corrected. The color description is corrected in the legend.

P.28, line 742: Fig 3: A montage of the social network composition of all participants with respect to healthy habits around the participant. Red dots are persons in the network with a negative health influence...

5. In Figure 3, it looks like that about ~25% of all ego’s networks have healthy habits. I think is worth mentioning the exact percentage. Also, is the size of the network correlated (at least weakly) with how “unhealthy” is the network? It should be that the more the friends I have the largest the probability that at least one of them has unhealthy habits. Therefore, the way networks are ranked in Figure 3 is related to the way that networks in Figure 4 are ranked and visualized.

We found that 17% of participants had networks in which all persons were healthy. We added this result in the text. There is a weak negative correlation between network size and proportion of unhealthy network members (Pearson’s correlation = -0.13, 95% CI (-0.08 – -0.17)). We added this result to the text.

P.7, line 153: 17% of participants had personal networks in which all members were healthy....There was a weak negative correlation between network size and the percentage of network members with unhealthy habits (Pearson’s correlation = -0.13, 95% confidence interval (-0.08 – -0.17)).

6. My above comment on the correlation between network size and percentage of friends that follow unhealthy habits is the main reason (from my point of view) behind the result that the authors report in Table 3, i.e. the association between the MSRS and the size of the network. There is this confounding here that the structure (at least the size) is highly correlated with the network composition - health. This is something that needs to be discussed in the paper – maybe in page 11.

We had the same concern, and addressed it by dividing all compositional trait frequencies by network size to create the percentage variables. Therefore, the percentage of persons who don’t exercise are the number of network members who don’t exercise divided by network size. Therefore, we believe our analyses account for size. We emphasize this on page 5.

P.5, line 110: All compositional variables were created to account for network size. Specifically, the number who fit a category were divided by the total size to create the percentage.

7. At lines 232-247 the authors discuss about “homophily”. It is a common sense that association (correlation) of health behaviors in a social network can be attributed to homophily (the tendency of similar individuals to connect in a social network) or to contagion effects, or to the fact that certain areas of a network subject to same externalities. In line 234 the authors state that a “potential mechanism of homophily is social contagion.” I would say that potential mechanism of the association in health behaviors are homophily and social contagion.

We changed this wording throughout the paragraph to not overlap with the prior paragraph that already discusses homophily and instead focus on social contagion and externalities.

P.11, line 256: Two more mechanisms that may explain the association of network members' health habits and the participant's neurological disability are social contagion and antecedent exposures. Social contagion is a type of social influence in which behavior in one or many network members affects the behavior of the index participant. Detection of this effect requires longitudinal data and network modeling, such as stochastic actor-oriented or instrumental variable approaches, to understand the spread of behaviors through social ties. For example, one study shows the spread of physical activity in 1 million users of a smartphone running app³¹. Antecedent exposures influencing both parties may be another contributor. For example, rural environments with poor access to medical services may influence the habits of all members of the network with regard to seeking medical care. Finally, a combination of these factors may explain the association of poor health habits in the network and a person's neurological disability.

Reviewer 2's comments:

This is a very well written and interesting paper that looks at a novel metric that may impact on health. As such it provides new information that may be used to influence the risk of developing MS in those at risk. The social network is certainly a very topical area for research and the authors address this in the GEMS cohort.

I have several questions.

1. Firstly the MSRS_R was used as the primary outcome measure of disability but was heavily skewed to a zero response. Did the authors consider using a hurdle model to analyze the data?

We completed a hurdle model with both poisson regression and negative binomial regression to analyze the data. We found they were similar to our non-parametric spearman rank associations (see Supplementary Table 1). The spearman rank associations are robust to non-normal distributions and are easy to interpret. Therefore, after considering all the models, we kept the non-parametric spearman rank associations as the main sensitivity analysis.

2. What was the influence of educational level on social networks could this be a confounder?

We agree that years of education could be a confounder. Therefore, we completed additional analyses and revised our linear regression analyses now adjusting for age, sex, marital status, and years of education. The two strongest network composition variables, percent of network members (1) who do not go to the doctor, or (2) are deemed to have a negative health influence on the respondent, remain significant. We modified Table 4 and the text on P.8 to reflect the revised results.

P.8, line 175: To deconstruct these global effects of the social network, we examined the association of individual network metrics with the MSRS-R, adjusting for sex, age, marital status, and years of education (**Table 4**). None of the network structure metrics was significantly associated with MSRS-R score, consistent with the global assessment. Two network composition features were significantly associated with MSRS-R score: the percent of network members who (1) do not go to a doctor regularly, or (2) are deemed to have a negative health influence on the respondent. The strongest association was with the percent of network members who are deemed to have a negative health influence ($\beta=0.017\pm 0.005$, $p=0.016$).

3. In table 2 it would be useful to provide the actual percentages who smoke etc in addition to the medians IQR.

This is corrected to show the actual percentages with both a median value and the percentiles for the health habits data.

4. Could the authors also comment on how representative their sample was of the whole GEMS cohort would it be more likely that those who respond have a greater or lesser social network than those who did not?

The study population in this manuscript is representative of the overall cohort with respect to demographic profile. In this study, we sent out the social network questionnaire to all participants (as of October 2016) and set a data collection deadline 5 weeks later. We expect the participants who completed the social network questionnaire to be more engaged with the research study than those who did not. However, it is also possible that some participants were simply unavailable to complete the questionnaire during the time frame. It is very difficult to say whether those who completed the questionnaire have a different social network structure or composition than those who did not complete the questionnaire, simply because we do not have data from the latter.

1 August 14, 2018

2 **A Scalable Online Tool for Quantitative Social Network**
3 **Assessment Reveals Potentially Modifiable Social Environmental**
4 **Risks**

5 Amar Dhand^{1,2,*}, Charles C. White³, Catherine Johnson⁴, Zongqi Xia⁵, Philip De Jager⁴

6

7 ¹Department of Neurology, Brigham and Women's Hospital, Harvard Medical School,
8 Boston, USA

9 ²Network Science Institute, Northeastern University, Boston, USA.

10 ³Broad Institute, Program in Medical and Population Genetics, Cambridge, USA.

11 ⁴Multiple Sclerosis Center and the Center for Translational & Computational
12 Neuroimmunology, Department of Neurology, Columbia University Medical Center,
13 New York, USA.

14 ⁵Department of Neurology, University of Pittsburgh, Pittsburgh, USA.

15

16 To whom correspondence should be addressed:

17 *Amar Dhand, M.D., D.Phil.

18 Brigham and Women's Hospital

19 75 Francis Street

20 Boston, MA 02115

21 617-732-5330; fax: 61-732-6083; Email: adhand@bwh.harvard.edu

22 **ABSTRACT**

[revised manuscript text omitted]

517

518